# Spin transport in insulators without exchange stiffness

Koichi Oyanagi [1]*, Saburo Takahashi[1,2,3], Ludo J. Cornelissen[4], Juan Shan[4], Shunsuke Daimon[1,2,5], Takashi Kikkawa [1,2], Gerrit E.W. Bauer[1,2,3,4], Bart J. van Wees[4] & Eiji Saitoh[1,2,3,5,6]

The discovery of new materials that efficiently transmit spin currents has been important for spintronics and material science. The electric insulator $Gd_3Ga_5O_{12}$ (GGG), a standard substrate for growing magnetic films, can be a spin current generator, but has never been considered as a superior conduit for spin currents. Here we report spin current propagation in paramagnetic GGG over several microns. Surprisingly, spin transport persists up to temperatures of $100\,K \gg T_g = 180\,mK$, the magnetic glass-like transition temperature of GGG. At 5 K and 3.5 T, we find a spin diffusion length $\lambda_{GGG} = 1.8 \pm 0.2\ \mu m$ and a spin conductivity $\sigma_{GGG} = (7.3 \pm 0.3) \times 10^4\ Sm^{-1}$ that is larger than that of the record quality magnet $Y_3Fe_5O_{12}$ (YIG). We conclude that exchange stiffness is not required for efficient spin transport, which challenges conventional models and provides new material-design strategies for spintronic devices.

[1] Institute for Materials Research, Tohoku University, Sendai 980-8577, Japan. [2] Advanced Institute for Materials Research, Tohoku University, Sendai 980-8577, Japan. [3] Center for Spintronics Research Network, Tohoku University, Sendai 980-8577, Japan. [4] Physics of Nanodevices, Zernike Institute for Advanced Materials, University of Groningen, 9747 AG Groningen, The Netherlands. [5] Department of Applied Physics, The University of Tokyo, Tokyo 113-8656, Japan. [6] Advanced Science Research Center, Japan Atomic Energy Agency, Tokai 319-1195, Japan. *email: k.0yanagi444@gmail.com

According to conventional wisdom, spin currents can be carried by conduction electrons and spin waves[1–4]. Mobile electrons in a metal carry spin currents over distances typically less than a micron, whereas spin waves, the collective excitation of the magnetic order parameter[5–12] can communicate spin information over much longer distances (Fig. 1a). In particular, the ferrimagnetic insulator $Y_3Fe_5O_{12}$ (YIG) supports spin transport over up to a millimeter[1,2].

$Gd_3Ga_5O_{12}$ (GGG) is an excellent substrate material for the growth of, e.g., high-quality YIG films[13]. Above $T_g = 180$ mK, it is a paramagnetic insulator (band gap of 6 eV[14]) with $Gd^{3+}$ spin-7/2 local magnetic moments that are weakly coupled by an effective spin interaction[15] $J_{ex} \sim 100$ mK. Recently, the spin Seebeck effect (SSE), i.e., thermal spin current generation, was observed in GGG at low temperatures (< 20 K) and high magnetic fields[16].

Here, we report long-range (500 nm) spin transport in a GGG slab (Fig. 1b) under applied magnetic fields even at much higher temperature (~100 K) than the Curie–Weiss temperature $|\Theta_{CW}|$, whereas at low temperatures GGG turns out to be a surprisingly good spin-conductor.

## Results

**Material characterization.** GGG does not exhibit long-range magnetic ordering at all temperatures[15] (Fig. 1d), while its field-dependent magnetization is well described by the Brillouin function. The large low-temperature saturation magnetization of ~7 $\mu_B$ per $Gd^{3+}$ (Fig. 1c) is governed by the half-filled 4f-shell of the $Gd^{3+}$ local moments.

**Sample structure and measurement setup.** We adopt the standard nonlocal geometry[5–12] to study spin transport in a device comprised by two Pt wires separated by a distance $d$ on top of a GGG slab (Fig. 2b and Supplementary Note 1). Here, spin currents are injected and detected via the direct and inverse spin Hall effects[1,17] (SHE and ISHE), respectively (Fig. 2a). A charge current, $J_c$, in one Pt wire (injector) generates non-equilibrium spin accumulation $\mu_s$ with direction $\sigma_s$ at the Pt/GGG interface by the

SHE. When $\sigma_s$ and the magnetization **M** in GGG are collinear, the interface spin-exchange interaction transfers spin angular momentum from the conduction electron spins in Pt to the local moments in GGG at the interface, thereby creating a non-equilibrium magnetization in the GGG beneath the contact that generates a spin diffusion current into the paramagnet. Some of it will reach the other Pt contact (detector) and generate a transverse voltage in Pt by means of spin pumping into Pt and the ISHE.

To our knowledge, long-range spin transport in magnets has been observed only below their Curie temperatures[5–12], e.g., YIG, $NiFe_2O_4$, and $\alpha$-$Fe_2O_3$, so magnetic order and spin-wave stiffness have been considered indispensable. Here, we demonstrate that a relatively weak magnetic field can be a sufficient condition for efficient spin transport, demonstrating that the dipolar interactions alone generate coherent spin waves.

**Observation of long-range spin transport through paramagnetic insulator.** First, we discuss the field and temperature dependence of the nonlocal detector voltage $V$ in Pt/GGG/Pt with contacts at a distance $d = 0.5$ μm. We use the standard lock-in technique to rule out thermal effects (see Methods). A magnetic field $B$ is applied at angle $\theta = 0$ in the $z$–$y$ plane (see Fig. 2b) such that the magnetization in GGG is parallel to the spin polarization $\sigma_s$ of the SHE-induced spin accumulation in the injector.

Figure 2c shows $V(B)$ at 300 and 5 K. Surprising is the voltage observed at low, but not ultralow temperatures that increases monotonically as a function of $|B|$ and saturates at about 4 T. Pt/YAG/Pt, where YAG is the diamagnetic insulator $Y_3Al_5O_{12}$, does not generate such a signal (Fig. 3b); apparently the paramagnetism of GGG (Fig. 1c) is instrumental in the effect.

We present the nonlocal voltage in Pt/GGG/Pt at 5 K as a function of the out-of-plane magnetic field angle $\theta$ and injection-current $J_c$ as defined in Fig. 2b. The left panel of Fig. 2d shows that $V$ at $|B| = 3.5$ T is described by $V(\theta) = V_{max}\cos^2\theta$: it is maximal ($V_{max}$) at $\theta = 0$ and $\theta = \pm180°$ (**B** // **y**) but vanishes at $\theta = \pm90°$ (**B** // **z**) (also see Supplementary Note 2). Furthermore,

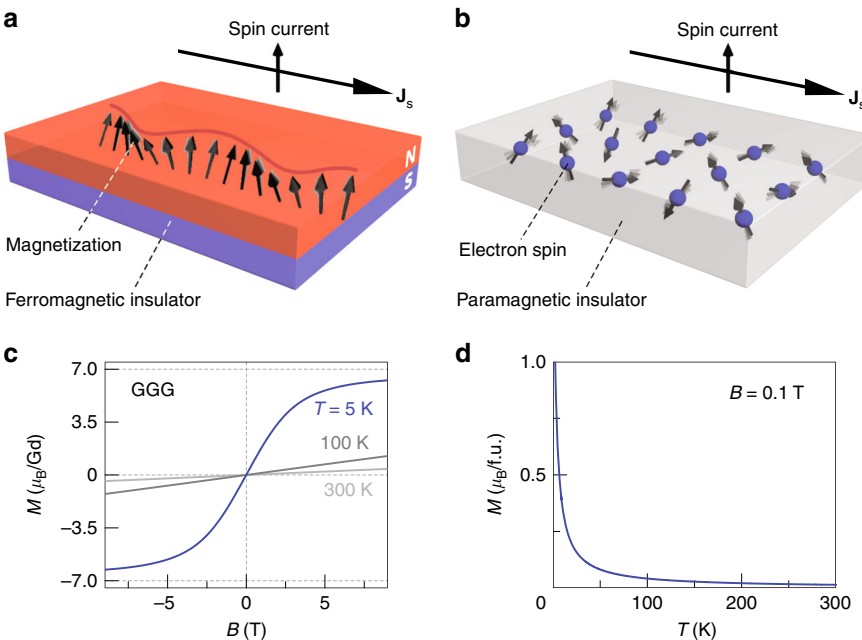

**Fig. 1** Concepts of spin current in a ferromagnetic insulator and a paramagnetic insulator and paramagnetism of $Gd_3Ga_5O_{12}$. **a** A schematic illustration of a ferromagnet, in which spins are aligned to form long-range order owing to strong exchange interaction. **b** A schematic illustration of a paramagnet, in which directions of localized spins are random due to thermal fluctuations. **c** Magnetization $M$ as a function of the applied magnetic field $B$ at 5 K, 100 K, and 300 K. The saturation magnetization of GGG is ~7 $\mu_B$ per $Gd^{3+}$ at 5 K. **d** The temperature dependence of the magnetization of GGG at $B = 0.1$ T

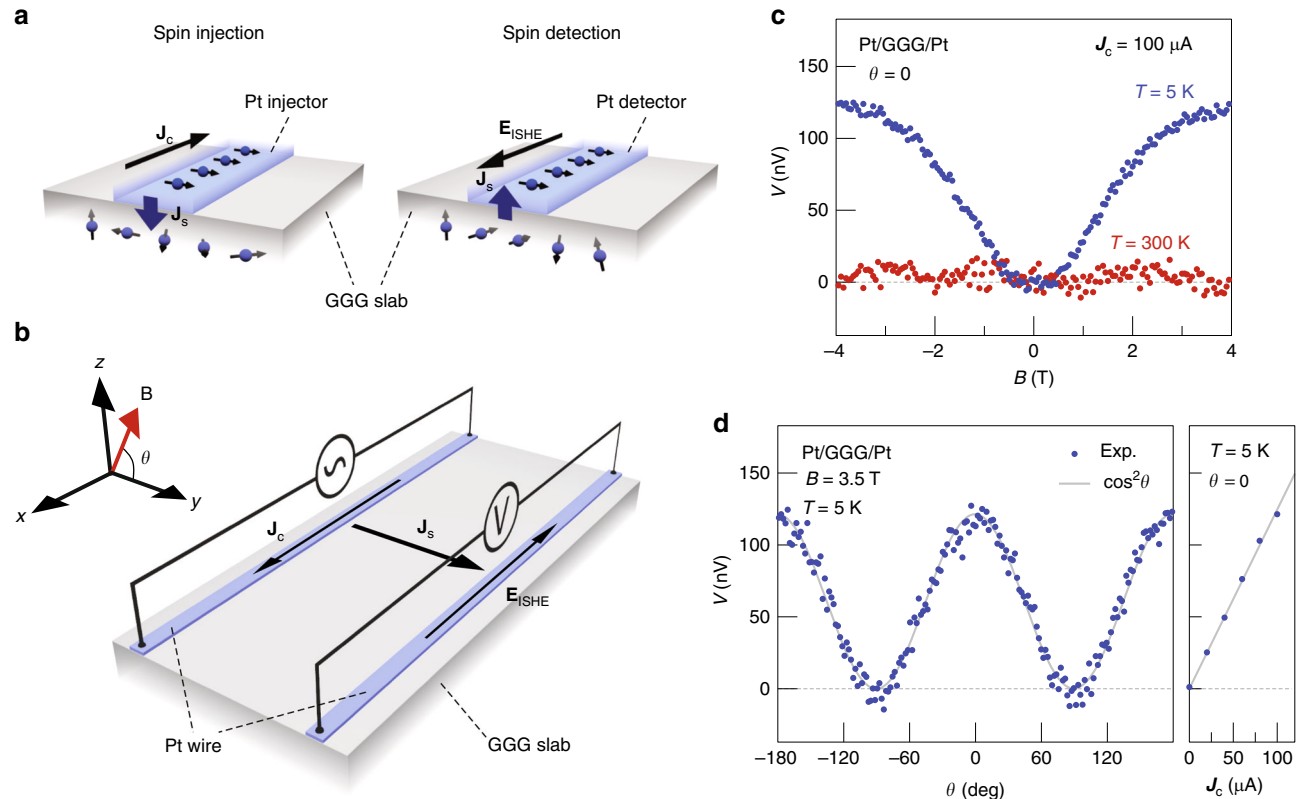

**Fig. 2** Observation of long-range spin transport through a paramagnetic insulator. **a** Schematics of spin injection (left panel) and detection (right panel) at two Pt/GGG contacts. $J_c$ and $J_s$ denote the spatial directions of charge and spin currents, respectively. $J_s$ is injected into GGG by applying $J_c$ via the SHE in Pt. At the detector, $J_s$ is driven in the direction normal to the interface and is converted into $J_c$ via the ISHE in Pt. **b** A schematic of the experimental setups. The nonlocal device consists of two Pt wires patterned on a GGG slab. **B** and $\mathbf{E}_{ISHE}$ denote the directions of the applied magnetic field and the electric field induced by the ISHE, respectively. We apply $J_c$ to the left Pt wire and detect the voltage $L|\mathbf{E}_{ISHE}|$ between the ends of the right Pt wire with length $L$. **c** The $B$ dependence of $V$ at $\theta = 0$ for Pt contacts separated by $d = 0.5\,\mu m$ at 300 K (red plots) and 5 K (blue plots) for $|B| < 4$ T. **d** The $\theta$ and $J_c$ dependence of $V$ for the same device at 5 K. The left panel shows the $\theta$ dependence of $V$ while $B = 3.5$ T was rotated in the $z$-$y$ plane; the gray line is a $\cos^2\theta$ fit. The right panel shows the $J_c$ dependence of $V_{max}$, determined by fitting $V_{max}\cos^2\theta$ to the $\theta$ dependence. We subtracted a constant offset voltage $V_{offset}$ from $V$ in **c** and **d** (see Supplementary Note 2)

$V$ depends linearly on $|J_c|$ (see the right panel of Fig. 2d). The same $\cos^2\theta$ dependence in Pt/YIG/Pt is known to be caused by the injection and detection efficiencies of the magnon spin current by the SHE and ISHE, respectively, that both scale like $\cos\theta$ (refs. [5–7]). The SHE-induced spin accumulation at the Pt injector interface, i.e., the driving force for the nonlocal magnon transport, scales linearly with $J_c$, and so does the nonlocal $V$ (ref. [5]). The above observations are strong evidence for spin current transport in paramagnetic GGG without long-range magnetic order.

**Temperature and high-field dependence of nonlocal spin signal.** Figure 3d shows the $\theta$ dependence of $V$ at $B = 3.5$ T and various temperatures $T$. Clearly, $V \sim V_{max}(T)\cos^2\theta$, where $V_{max}(T)$ in Fig. 3a decreases monotonically for $T > 5$ K, nearly proportional to $M(T)$ at the same $B$ as shown in the inset to Fig. 3a. The field-induced paramagnetism therefore has an important role in the $|V|$ generation. Surprisingly, $V_{max}$ at 3.5 T persists even at 100 K, which is two orders of magnitude larger than $|\Theta_{CW}| = 2$ K. The exchange interaction at those temperatures can therefore not play any role in the voltage generation. In contrast, a large paramagnetic magnetization (~$\mu_B$ per f.u. at 3.5 T) is still observed at 100 K, consistent with long-range spin transport carried by the field-induced paramagnetism.

At high fields, Fig. 3e shows a non-monotonic $V(B, \theta = 0)$: At $T < 30$ K, $V$ gradually decreases with field after a maximum at

~4 T, which becomes more prominent with decreasing $T$. A similar feature has been reported in Pt/YIG/Pt[9] and interpreted in terms of the freeze-out of magnons: a Zeeman gap $\propto B$ larger than the thermal energy $\propto T$ critically reduces the magnon number and conductivity. It appears that thermal activation of magnetic fluctuations is required to enable a spin current in GGG as well.

**Estimation of spin diffusion length.** By changing the distance $d$ between the Pt contacts, we can measure the penetration depth of an injected spin current. $V_{max}$ at 5 K, as plotted in Fig. 4a, decreases monotonously with increasing $d$. A similar dependence in Pt/YIG/Pt is well described by a magnon diffusion model[18,19]. We postulate that the observed spin transport in GGG can be described in terms of magnon diffusion of purely dipolar spin waves. As the GGG thickness of 500 μm $\gg d$, we cannot use a simple one-dimensional diffusion model that would predict a simple exponential decay $V_{max}(d) \sim \exp(-d/\lambda)$. Considering two spatial dimensions (see Supplementary Note 4) leads to:

$$V_{max}(d) = CK_0(d/\lambda) \tag{1}$$

where $K_0(d/\lambda)$ is the modified Bessel function of the second kind, $\lambda = \sqrt{D\tau}$ is the spin diffusion (relaxation) length, $D$ is the spin diffusion constant, $\tau$ is the spin relaxation time, and $C$ is a numerical coefficient that does not depend on $d$. By fitting Eq. (1) to the experimental data, we obtained $\lambda_{GGG} = 1.82 \pm 0.19\,\mu m$ at $B = 3.5$ T and $T = 5$ K.

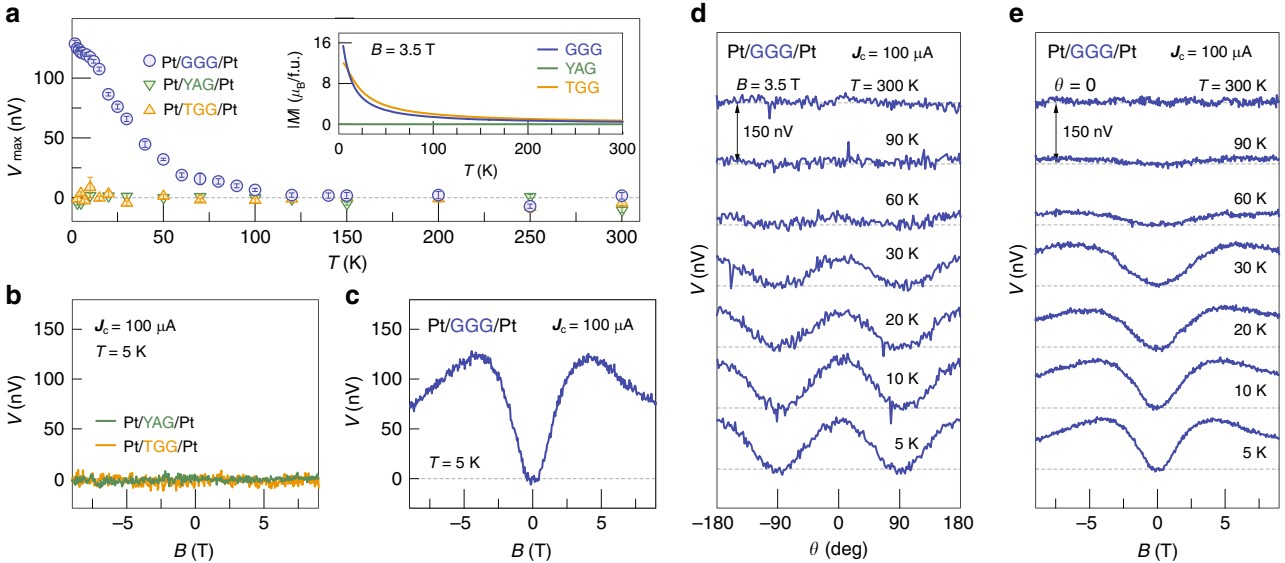

**Fig. 3** Temperature and magnetic field dependence of the nonlocal voltage signal. All experimental data were obtained by the same device ($d = 0.5\,\mu$m) with a current amplitude of 100 μA. **a** The temperature ($T$) dependence of the amplitude of the maximum nonlocal voltage $V_{max}$ for Pt/GGG/Pt, Pt/TGG/Pt, and Pt/YAG/Pt obtained from a sinusoidal fit to the magnetic field angle $\theta$ dependences of $V$ at $B = 3.5$ T. The error bars represent the 68% confidence level (±s.d.). The inset shows the $T$ dependence of the magnetization $M$ of GGG, TGG, and YAG at $B = 3.5$ T. **b, c** Comparison between $V$ for Pt/YAG/Pt, Pt/TGG/Pt, and Pt/GGG/Pt. **b** (**c**) shows the $B$ dependences of $V$ for Pt/YAG/Pt and Pt/TGG/Pt (Pt/GGG/Pt) at 5 K for $|B| < 9$ T. **d, e** The $\theta$ and $B$ dependence of $V$ for Pt/GGG/Pt at various temperatures. We varied $\theta$ by rotating the field at $B = 3.5$ T in the $z$–$y$ plane. The field was changed from $-9$ T to 9 T at $\theta = 0$ for the $B$ dependence

A long spin relaxation length in an insulator implies weak spin-lattice relaxation by spin-orbit interaction, which in GGG should be weak because the $4f$-shell in $Gd^{3+}$ is half-filled with zero orbital angular momentum ($L = 0$). We tested this scenario by a control experiment on a Pt/$Tb_3Ga_5O_{12}$ (TGG)/Pt nonlocal device with similar geometry ($d = 0.5\,\mu$m). TGG is also a paramagnetic insulator with large field-induced $M$ at low temperatures (see the inset to Fig. 3a). However, $Tb^{3+}$ ions have a finite orbital angular momentum ($L = 3$) and an electric quadrupole that strongly couples to the lattice[20]. Indeed, we could not observe any nonlocal voltage in Pt/TGG/Pt in the entire temperature range (see Fig. 3a–c). This result highlights the importance of a weak spin-lattice coupling in long-range paramagnetic spin transmission.

**Modeling of nonlocal spin signal.** We model the nonlocal voltages in a normal-metal (N)/paramagnetic insulator (PI)/normal-metal (N) system by the magnon diffusion equation in the PI and interface exchange interactions at the metal contacts[21] with spin-charge conversion (see Supplementary Notes 3 to 6). The voltage in the Pt detector as a function of $B$ and $T$ reads:

$$V(B, T) = C_1 \frac{g_s^2}{\sigma_{GGG}} \frac{[\xi B_S(2S\xi)/\sinh(\xi)]^2}{[1 + C_2 \frac{g_s}{\sigma_{GGG}} \xi B_S(2S\xi)]^2} \quad (2)$$

where $\xi(B, T) = g\mu_B B / k_B(T + |\Theta_{CW}|)$, $g$ is the $g$-factor, $\mu_B$ is the Bohr magneton, $k_B$ is the Boltzmann constant, $B_S(x)$ is the Brillouin function as a function of $x$ for spin-$S$, $C_1$ and $C_2$ are known numerical constants, $S = 7/2$ is the electron spin of a $Gd^{3+}$ ion, $g_s$ is the effective spin conductance of the Pt/GGG interface, and $\sigma_{GGG}$ is the spin conductivity in GGG. The observed $V(B)$ is well described by Eq. (2) (see Fig. 4b). The best fit of Eq. (2) is achieved by $\sigma_{GGG} = (7.25 \pm 0.26) \times 10^4$ $Sm^{-1}$ and $g_s = (1.82 \pm 0.05) \times 10^{11}$ $Sm^{-2}$. We determine $\sigma_{GGG}$ and $g_s$ by fitting the experimental data to a numerical (finite-element) simulation of the diffusion[19] that takes the finite width of the contacts and the GGG film into account (the details are discussed in Supplementary Note 9).

Surprisingly, the obtained $\sigma_{GGG}$ ($g_s$) value is at the same temperature eight (six) times larger than that of the Pt/YIG/Pt sample[8] ($\sigma_{YIG} = (0.9 \pm 0.6) \times 10^4$ $Sm^{-1}$ and $g_s = 0.3 \times 10^{11}$ $Sm^{-2}$ at 10 mT), which is evidence for highly efficient paramagnetic spin transport.

**Discussion**

The reported spin transport in GGG has not been observed nor does affect previous nonlocal experiments in YIG/GGG, conducted at room temperature[5,9] or at low temperatures and low fields[6–8], because under those conditions GGG is not magnetically active. Non-negligible contributions from GGG may appear only at low-temperature and high fields, as observed in the local SSE of a Pt/YIG/GGG junction[22].

A few papers address the spin current in paramagnets[16,23–26]. Shiomi et al. and Wu et al. reported paramagnetic spin pumping in $La_2NiMnO_6$ and SSE in GGG and $DyScO_3$ in their paramagnetic phases, respectively. Spin currents were observed above but close to the magnetic ordering temperature and therefore attributed to critical spin fluctuations (paramagnons). Here we find a nonlocal signal in GGG even at 100 K, much higher than $|\Theta_{CW}| = 2$ K in GGG (see Fig. 3a). The exchange interaction has been reported to cause spin diffusion in Heisenberg paramagnets[27] at zero magnetic fields. We cannot measure a possible contribution to the spin transport by diffusion at (nearly) zero magnetic fields in our setup by the spin conductivity mismatch and the decrease of the spin diffusion length at small field (see Supplementary Note 7 and 8). However, direct spin diffusion is suppressed when the rotational symmetry is broken by anisotropy or applied magnetic fields[28] (exponentially so in Heisenberg ferromagnets[29]). The contribution from direct magnetic dipole–dipole interactions dominates nuclear spin diffusion, but is orders of magnitude smaller than what we observe[30].

A plausible mechanism is based on the long-range dipolar interaction, which becomes important when the Zeeman and thermal energies are of comparable magnitude. The paramagnetic

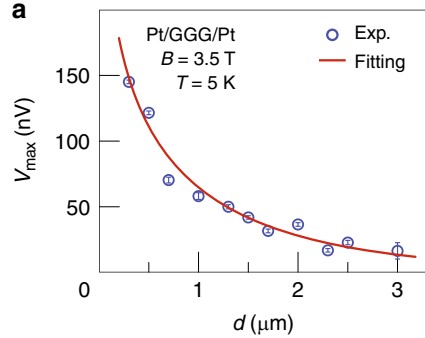
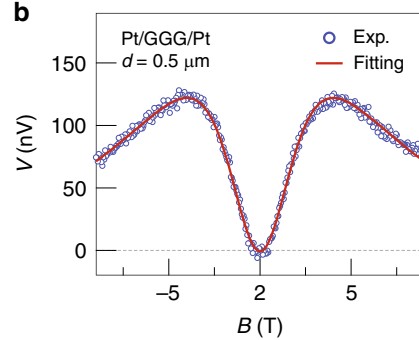

**Fig. 4** Comparison with theory. **a** The experimental results (blue circles) and calculations (solid red line) for $V_{max}$ in Pt/GGG/Pt as a function of the separation $d$ between the Pt contacts and an applied current of 100 μA. We obtain $V_{max}$ by a $\cos^2\theta$ fit to the magnetic field angle $\theta$ dependence of $V$ at $B = 3.5$ T and $T = 5$ K. By using Eq. (1), the spin diffusion length $\lambda_{GGG} = 1.82 \pm 0.19$ μm. The error bars represent the 68% confidence level (±s.d.). **b** The experimental results (blue circles) and calculation (solid red line) of the $B$ dependence of the nonlocal $V$ for Pt/GGG/Pt for $|B| < 9$ T. The experimental data shown in **b** are the same as those shown in Fig. 3c

system acquires a finite magnetization by applying magnetic field, and the dipole interaction is enhanced, especially in GGG with large Gd magnetic moments. The dipole interaction can support collective spin-wave excitations even in paramagnets that for small wave numbers are identical to those of ferromagnets (see Supplementary Discussion).

Finally, we address the large spin conductivity of GGG at low temperatures and high magnetic fields. Thermally occupied magnons with energies up to about $k_BT$ carry the spin transport. In YIG, their dispersion is dominated by the exchange interaction. Although large exchange energy corresponds to a large magnon group velocity, exchange magnons have short wave lengths, which make them sensitive to local magnetic disorder such as grain boundaries that give rise to spin-wave scattering limiting YIG's spin conductivity. In contrast, the dipolar interaction is long-ranged and therefore less affected by (short-range) disorder. In GGG, the long-range dipole interaction dominates the excitation of spin waves for frequencies comparable to the Zeeman energy ($\approx g\mu_B B$). Therefore, paramagnets with strong dipolar but weak exchange interactions, such as GGG, may provide ideal spin conductors as long as the thermal fluctuation of the magnetic moments is sufficiently suppressed by applied magnetic fields. Moreover, the large spin conductance $g_s$ indicates a strong interface exchange interaction at a metal contact to GGG.

In summary, we discovered spin transport in the Curie-like paramagnetic insulator $Gd_3Ga_5O_{12}$ over several microns. Its transport efficiency at moderately low temperatures and high magnetic fields is even higher than that of the best magnetically ordered material YIG. Low-temperature experiments[31], covering the magnetic glass-like transition temperature of GGG, may be interesting to explore the link between spin frustration and spin transport in GGG. Magnonic crystals[2] can help determining the length scales that govern the observed spin transport. Since paramagnetic insulators are free from Barkhausen noise associated to magnetic domains and magnetic after-effects (aging)[32] typical for ferromagnets and antiferromagnets, they are promising materials for future spintronics devices.

## Methods

**Sample preparation**. A single-crystalline $Gd_3Ga_5O_{12}$ (111), $Y_3Al_5O_{12}$ (111), and $Tb_3Ga_5O_{12}$ (111) (500 μm in thickness) were commercially obtained from CRYS-TAL GmbH, Surface Pro GmbH, and MTI Corporation, respectively. For magnetization measurements, the slabs were cut into 3 mm long and 2 mm wide. For transport measurements, a nonlocal device with Pt wires was fabricated on a top of each slab by an e-beam lithography and lift-off technique. Here, the Pt wires were deposited by magnetron sputtering in a $10^{-1}$ Pa Ar atmosphere. The dimension of the Pt wire is 200 μm long, 100 nm wide, and 10 nm thick and the separation distance between the injector and detector Pt wires are ranged from 0.3 to 3.0 μm.

A microscope image of a device is presented in Supplementary Fig. 1. We measured the temperature dependence of the resistance between the two Pt wires on the GGG substrate but found that the resistance of the GGG is too high to be measurable.

**Magnetization measurement**. The magnetization of GGG, YAG, and TGG slabs was measured using a vibrating sample magnetometer option of a quantum design physical properties measurement system (PPMS) in a temperature range from 5 K to 300 K under external magnetic fields up to 9 T.

**Spin transport measurements**. We measured the spin transport with a PPMS by a standard lock-in technique from 5 K to 300 K. An a.c. charge current was applied to the injector Pt wire with a Keithley 6221 and the voltage across the detector Pt wire was recorded with a lock-in amplifier (NF 5640). The typical a.c. charge current has a root-mean-square amplitude of 100 μA and a frequency of 3.423 Hz.

## Data availability

The data that support the findings of this study are available from the corresponding author on request.

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

## Acknowledgements

We thank S. Maekawa, J. Barker, R. Iguchi, T. Niizeki, and D. Hirobe for discussions, and R. Yahiro for experimental help. This work is a part of the research program of ERATO Spin Quantum Rectification Project (No. JPMJER1402) from JST, the Grant-in-Aid for Scientific Research on Innovative Area Nano Spin Conversion Science (Nos. JP26103005 and JP26103006), the Grant-in-Aid for Scientific Research (S) (No. JP19H05600), Grant-in-Aid for Research Activity Start-up (Nos. JP18H05841 and JP18H05845), 19H006450 from JSPS KAKENHI, JSPS Core-to-Core program the International Research Center for New-Concept Spintronics Devices, World Premier International Research Center Initiative (WPI) from MEXT, Japan, Netherlands Organization for Scientific Research (NWO), and NanoLab NL. K.O. acknowledges support from GP-Spin at Tohoku University.

## Author contributions

K.O. designed the experiment, fabricated the samples, collected all of the data. S.T. formulated the theoretical model. K.O. and S.T. analyzed the data. S.T. and K.O. estimated the parameters. L.J.C. and J.S. carried out the numerical simulation. T.K., B.J.v.W., G.E.W.B., and E.S. developed the explanation of the experimental results. E.S. supervised the project. K.O., S.T., L.J.C., J.S., S.D., T.K., G.E.W.B, B.J.v.W, and E.S. discussed the results and commented on the manuscript.

## Competing interests

The authors declare no competing interests.
