## [Peer Review File · Nature Communications]

Reviewers' comments:

Reviewer #1 (Remarks to the Author):

The paper reports on an interesting experimental finding, namely spin current generation and propagation in a paramagnetic Gd₃Ga₅O₁₂ (GGG) sample. The current persists up to 100 K. That GGG supports spin current is known from previous works (such as Wu et al Phys. Rev. Lett. 114, 186602 (2015) which was to my knowledge the first experimental work followed by others). Thus it seems a bit overstated and/or puzzling to claim "GGG ... has never been considered as a conduit for spin currents." or "Paramagnetic insulators have not attracted the spintronics community's attention since they seem unlikely carriers of spin currents." while previous evidence of this fact already exists. Also priority claims made in the conclusion part "In summary, we discovered long-range spin transport in the Curie-like paramagnetic..." need to be put in the right perspective.

To the physics:

1. In my view the presented phenomenological theory is an extension/application of the ferromagnetic case and is rather descriptive in nature. What is the role of the detection process?. Is it clear at all that one can simply make the same assumptions as in the ferromagnetic case. Is there any ab-initio theory justification or experimental evidence that Pt/YIG couples in the same way as Pt/GGG.
2. With the arguments presented in the main paper regarding the role of the dipolar interaction one may wonder whether the observations are sample-quality/sample-geometry or sample-size dependent. How does the ISHE-voltage depends on the width/quality of the Pt stripe.
3. Considering the works on spin currents in YIG/GGG, is there a spin current flow across the whole system and how do both components interplay.

In summary, though the experiments are indeed interesting, from the materials at hand, I do not see a compelling evidence for breakthrough findings nor for new fundamental theory insight.

Reviewer #2 (Remarks to the Author):

The authors have reported nonlocal spin transport in paramagnetic insulator GGG over several micrometers. They demonstrated spin current injection and detection via SHE and ISHE by using two separated non-magnetic Pt wires on top of the GGG. The spin transport can persist up to temperatures of 100 K, much higher than the GGG's magnetic glass-like transition temperature. The paper is well-written and the results are interesting. Paramagnetic spin pumping from a ferromagnetic material above the Curie temperature was first demonstrated in PRL 113 266602 (2014). Then the paramagnetic spin Seebeck effect in GGG and DSO was also reported in PRL 114 186602 (2015). Considering the reported micro voltages of the spin Seebeck results of GGG in 5 K ~ 20 K in the second paper, the measurement of the hundreds of nanovolts nonlocal spin transport can be expected and reasonable. However, the explanation of the spin transport is completely different between the nonlocal measurements here and the SSE reported previous (PRL 114 186602

(2015)). The most important issue in this manuscript is that the authors should comment on explanations of the SSE studies and clarify clearly the differences between the physics explanations before publishing it. Also the magnon descriptions of this work is different from the very similar studies on YIG (Nat Phys 11,1022-1026 (2015)) and the value of magnon frequencies should be given and explained. The detailed concerns are as follows:

Spin fluctuations or paramagnons have been extensively reported to exist up to temperatures well above T_c or T_N (Phys Rev 180 591 (1969), JAP 55, 1869 (1984), even $T/T_c=10$ in PRB 31 5884 (1985)) in systems with existence of short-range magnetic interactions. The authors should compare the difference of underlying physics and clarify why they gave completely different explanations (long-range dipolar interactions).

In the discussion part, the authors claim that only low-frequency spin waves close to $k=0$ contribute to the spin transport. What are the values of the frequencies? The magnons are quantized spin waves with $hf \approx k_B T$ in Nat Phys 11,1022-1026 (2015). Considering the identical nonlocal excitation and detection method with Pt wires, what is difference between the spin waves in this reference and in the authors' work?

Field dependence of the spin diffusion length should be presented. For YIG, the spin diffusion length decreases for increasing the magnetic field (Figure 5a, PRB 93, 020403(R) (2016)). How about GGG in this work? The authors can use the same method to get the spin diffusion lengths for different magnetic fields easily.

In the abstract the authors said that at 5K spin conductivity is larger than YIG. That only happens when a large magnetic field is applied. So external magnetic field (4T?) should be mentioned in the abstract, or it will be a claim that is easily misunderstood to the readers.

Why the spin conductivity of GGG in that condition (5K, 4T) is better than in YIG?

Also the authors compared their spin conductivity results of GGG to the values in PRB 94, 180402(R) (2016) where the thickness of YIG is 210nm, with 500 μ m GGG on the bottom. Do the authors think the GGG substrate of YIG plays a role in the transport?

The authors emphasize that the spin transport in GGG persists up to temperatures much higher than GGG's glass-like transition temperature. What are the expectations of the authors for the GGG spin transport measurements around or below the spin glass temperature?

What is the difference of the experiment data between figure 2d, figure 3b right panel, figure 3d 5 K and figure 4b?

Figure 2 caption " B and E_{ISHE} denote the directions of the applied magnetic field and the electric field induced by the ISHE, respectively". E_{ISHE} is not labeled in Figure 2b.

Reviewer #3 (Remarks to the Author):

This manuscript reports the highly efficient spin transport in the paramagnetic insulator Gd₃Ga₅O₁₂ (GGG). By using the nonlocal geometry of Pt/GGG/Pt, the authors have found magnon transport in range of micrometer scale at temperatures far above the magnetic glass-like transition temperature.

They have determined the characteristic spin transport parameters. Remarkably, at 5K the spin diffusion length reaches 1.8 micrometers; and spin conductivity is even much larger than that of the best YIG, so is the spin mixing conductance at the interface. The authors have proposed that the spin transport mechanism in GGG is mainly due to the long-range dipolar interaction, whereas the magnon excitation in this paramagnetic insulator is caused by the process of interfacial spin-flip. The results are quite interesting in physics. I recommend its acceptance by Nature Communications. But the authors may answer the following remarks to clarify some issues.

1. The earlier nonlocal magnon spin transport experiments are mainly based on the structure of GGG (substrate)/YIG/Pt. Since the authors have claimed that the spin conductivity in GGG is even larger than that of YIG at low temperature (e.g. 5 K), is the early results about YIG spin conductivity and spin diffusion length (e.g. ref [10]) possibly contaminated because of the contribution of GGG at low temperature? It is better for the authors to give some comments about this issue.
2. The authors had better show the second-harmonic signal results, which are often considered in dealing with paramagnetic spin Seebeck effect. Magnon can be excited in magnetic insulators in a variety of ways, such as thermal gradient or low-frequency currents in heavy metals like Pt. It was reported that paramagnons and short range magnetic interaction are important in the thermal induced magnon excitations and diffusion in GGG (ref [17]), which seems contrary to the present conclusion that long-range dipolar interaction plays an important role. The authors should give some comment on this issue.

Reviewer #1 (Remarks to the Author):

The paper reports on an interesting experimental finding, namely spin current generation and propagation in a paramagnetic Gd₃Ga₅O₁₂ (GGG) sample. The current persists up to 100 K. That GGG supports spin current is known from previous works (such as Wu et al Phys. Rev. Lett. 114, 186602 (2015) which was to my knowledge the first experimental work followed by others). Thus it seems a bit overstated and/or puzzling to claim "GGG ... has never been considered as a conduit for spin currents." or "Paramagnetic insulators have not attracted the spintronics community's attention since they seem unlikely carriers of spin currents." while previous evidence of this fact already exists. Also priority claims made in the conclusion part "In summary, we discovered long-range spin transport in the Curie-like paramagnetic..." need to be put in the right perspective.

We thank the reviewer for sharing his/her insights and finding our experimental result "interesting". We apologize for misleading statements in the previous version of the manuscript that we removed or rephrased. Nevertheless, we would like to emphasize that we are the first to observe the long-range spin-current propagation in the paramagnetic insulator GGG. Previous work focused on the demonstration of spin-current generation in magnetic systems just above the critical temperatures, attributing the results to short-range exchange interactions (short-lived paramagnons) mediated by the critical spin fluctuations. Nobody could reasonably anticipate from those results that spin currents can flow over long distances in the paramagnetic phase at temperature well above ordering phase transitions. To our own surprise we found long-range spin transport in GGG over several hundred nanometers that persists up to 100 K, two orders of magnitude larger than the Curie-Weiss temperature of GGG ($|\theta_{CW}| = 2$ K), upsetting conventional wisdom. The physics behind our observation is necessarily different from previously observed phenomena, since thermal fluctuation mask weak exchange interactions at such high temperatures. We therefore attribute our discovery to the long-range dipolar interaction that is robust even at elevated temperatures when sufficiently strong magnetic fields are applied. We are confident that our work provides new insights into the physics of paramagnets and warrants

publication in Nature Communications.

To the physics:

1. In my view the presented phenomenological theory is an extension/application of the ferromagnetic case and is rather descriptive in nature. What is the role of the detection process?. Is it clear at all that one can simply make the same assumptions as in the ferromagnetic case. Is there any ab-initio theory justification or experimental evidence that Pt/YIG couples in the same way as Pt/GGG.

We thank the reviewer for bringing up the new and fundamental point of the spin injection/detection at the Pt/GGG interface.

At the normal-metal/insulator interface, the conduction electron spin σ in the metal exchange-couples only to the localized spins \mathbf{S} in the outermost layers of the insulator, as was shown by ab-initio theory [Jia *et al.*, Europhys. Lett. **96**, 17005 (2011)]. The spin mixing conductance is a function of σ and proportional to the number of the local interface moments $\langle \mathbf{S} \rangle$. In this respect, there is no fundamental difference between YIG and GGG.

To clarify this point, we now cite [X. Jia *et al.*, Europhys. Lett. **96**, 17005 (2011)] in Section C of the SI.

2. With the arguments presented in the main paper regarding the role of the dipolar interaction one may wonder whether the observations are sample-quality/sample-geometry or sample-size dependent. How does the ISHE-voltage depends on the width/quality of the Pt stripe.

As in other nonlocal experiments, the widths of the Pt contacts play a role when comparable to their distance. The magnetic film thickness dependence is not yet fully understood even for YIG. Otherwise, we do not expect nor observe sample-size and/or sample-geometry dependent features in the nonlocal voltage. Indeed, we can try to modify spin transport in GGG, e.g. by periodic surface structures with length scales

comparable to the dipolar spin wavelength. This is a hot topic in spintronics and magnonics, and its application to GGG is a promising research direction. However, we hope that the referee agrees that this topic is outside the scope of this work.

We added a sentence to describe the future perspective in the conclusion paragraph (Lines 173 - 174, Page 12).

3. Considering the works on spin currents in YIG/GGG, is there a spin current flow across the whole system and how do both components interplay.

We thank the reviewer for raising a fundamental question that is relevant for much of the spintronics with magnetic insulators. Indeed, based on our result, there should be a finite spin-current flow across a YIG/GGG interfaces at low temperatures ($T \lesssim 20$ K) and high magnetic fields ($B \sim$ several Tesla), where GGG's spin conductivity is large, comparable to, or even better than YIG. GGG then becomes a spin sink that reduces the spin accumulation at the injector and detector Pt/YIG interfaces. However, the interpretation of previous nonlocal 1ω experiments on Pt/YIG/GGG remains to be valid, since they were conducted typically at room temperature or for low- T in the low- B regime, in which the spin conductivity of GGG is negligibly small.

In this context we would like to point out the longitudinal spin Seebeck effect (LSSE) in Pt/YIG/GGG system at low- T and high- B [Phys. Rev. Lett. **117**, 207203 (2016)] in which the temperature gradient extends down to the YIG/GGG interfaces. In Fig. R1, we plot the magnetic field B (< 8 T) dependence of the normalized LSSE voltage S for a Pt/YIG/GGG [red solid line, Phys. Rev. Lett. **117**, 207203 (2016)] and a Pt/YIG-slab bilayer (without GGG) [blue solid line, Phys. Rev. B **92**, 064413 (2015)] at $T = 5$ K. The signal in Pt/YIG/GGG gradually *increases* with B (the sharp magnon polaron peak at 2.6 T is irrelevant for the present discussion). In sharp contrast, the Pt/YIG bilayer signal decreases with increasing B , simply by the field-induced freeze-out of magnons. The magnetic field effect in Pt/YIG/GGG becomes significant for $T \leq 20$ K [Phys. Rev. Lett. **117**, 207203 (2016)] while its B dependence is consistent with the paramagnetic LSSE in the Pt/GGG system reported by Wu *et al.* [Phys. Rev. Lett. **114**, 186602 (2015)]. We therefore attribute these observations to the thermal spin-current injection from GGG into YIG.

In the revised manuscript, we added a new paragraph to discuss the effect of spin-current flow at YIG/GGG interfaces, in Lines 130 - 134, Page 10.

Figure R1: Comparison of LSSE between Pt/YIG/GGG and Pt/YIG bilayer (without GGG). The red and blue solid lines represent the magnetic field B dependence of the normalized spin Seebeck voltage $S = E_{\text{LSSE}}/\nabla T$ in a Pt/YIG-film(4 μm)/GGG and a Pt/YIG-bulk(1 mm) at $T = 5$ K, respectively. E_{LSSE} is the measured electric field (voltage divided by the length of Pt), ∇T is the applied temperature gradient, and the numbers in parentheses represent the YIG thickness.

In summary, though the experiments are indeed interesting, from the materials at hand, I do not see a compelling evidence for breakthrough findings nor for new fundamental theory insight.

We again thank the reviewer for thoughtful input and finding our experiments interesting. The previous version of the manuscript has obviously not properly highlighted the novelty of our work, i.e. the long-range spin transport in paramagnets far above any critical temperature which, to the best of our knowledge, has never been reported before. The long-range transmission of spin information is a cornerstone in spintronics since essential for efficient spin-current circuits. Furthermore, paramagnetic insulators have clear advantages compared to conventional ferromagnets (e.g., absence

of magnetic-domains, Barkhausen noise, aging (magnetic after-effects), and magnetic anisotropy).

We are confident that the revised manuscript will introduce a broader audience to the unexpected physics of paramagnets and their applicability in spintronics that deserves publication in Nature Communications. We hope that Reviewer #1 can be swayed by our arguments to reconsider his/her recommendation.

Reviewer #2 (Remarks to the Author):

The authors have reported nonlocal spin transport in paramagnetic insulator GGG over several micrometers. They demonstrated spin current injection and detection via SHE and ISHE by using two separated non-magnetic Pt wires on top of the GGG. The spin transport can persist up to temperatures of 100 K, much higher than the GGG's magnetic glass-like transition temperature. The paper is well-written and the results are interesting. Paramagnetic spin pumping from a ferromagnetic material above the Curie temperature was first demonstrated in PRL 113 266602 (2014). Then the paramagnetic spin Seebeck effect in GGG and DSO was also reported in PRL 114 186602 (2015). Considering the reported micro voltages of the spin Seebeck results of GGG in 5 K \sim 20 K in the second paper, the measurement of the hundreds of nanovolts nonlocal spin transport can be expected and reasonable. However, the explanation of the spin transport is completely different between the nonlocal measurements here and the SSE reported previous (PRL 114 186602 (2015)). The most important issue in this manuscript is that the authors should comment on explanations of the SSE studies and clarify clearly the differences between the physics explanations before publishing it. Also the magnon descriptions of this work is different from the very similar studies on YIG (Nat Phys 11,1022-1026 (2015)) and the value of magnon frequencies should be given and explained. The detailed concerns are as follows:

We thank the reviewer for his/her positive assessment of our manuscript and for sharing her/his concerns about the physics behind our findings that we address in the following.

1. Spin fluctuations or paramagnons have been extensively reported to exist up to temperatures well above T_c or T_N (Phys Rev 180 591 (1969), JAP 55, 1869 (1984), even $T/T_c=10$ in PRB 31 5884 (1985)) in systems with existence of short-range magnetic interactions. The authors should compare the difference of underlying physics and clarify why they gave completely different explanations (long-range dipolar interactions).

We thank the reviewer for giving us the opportunity to highlight this important point in the revised manuscript.

First, we would like to point out the salient difference between the nonlocal SSE and the spin injection-detection experiments in ferromagnets. The former is caused by temperature gradients not only at the surface but also in the bulk of the magnet. We have shown previously that the competition between spin generation and diffusion can change the sign of the nonlocal SSE signal. Here we focus on spin injection at the interface by spin accumulation, which leads to spin signals that are easier to interpret.

That said, an essential difference between previous reports of the paramagnetic spin Seebeck effect (SSE) and our nonlocal results is their onset temperatures, which forced us to introduce a different scenario, viz. the magnetic dipole interaction, as elaborated below.

The “paramagnon” is a critical spin- fluctuation closely above the critical temperature T_c and precursor of long-range magnon formation. The paramagnetic SSE [Phys. Rev. Lett. **114**, 186602 (2015)] was also observed just above T_c ($T / T_c < 10$), and, hence, attributed to these paramagnons. Our results, on the other hand, cannot be explained by critical spin fluctuations. The onset temperature of our nonlocal signals is two orders of magnitude higher than the critical temperature of GGG, i.e. $T / T_c = 100$. At such a high temperatures, the thermal fluctuations overwhelm critical spin correlations, which rules out a paramagnon scenario. In our opinion the only remaining mechanism requires dipolar spin wave formation. Even at 100 K, GGG acquires a large field-induced magnetization ($\sim 1 \mu_B/\text{f.u.}$ at 3.5 T) with corresponding dipolar interaction that generates magnetic rigidity with collective excitations. In contrast to coherent spin waves (see our reply to Query 5), short-lived paramagnons cannot explain the giant spin conductivity of GGG either.

To clarify this point, we completely revised the discussion part (Lines 135 - 148, Pages 10 - 11), addressing the paramagnon scenario [Phys. Rev. Lett. **114**, 186602 (2015)] and the novelty of the present results in detail.

2. In the discussion part, the authors claim that only low-frequency spin waves close to $k=0$ can contribute to the spin transport. What are the values of the

frequencies? The magnons are quantized spin waves with $hf \approx k_B T$ in Nat Phys 11,1022-1026 (2015). Considering the identical nonlocal excitation and detection method with Pt wires, what is difference between the spin waves in this reference and in the authors' work?

We agree that this issue deserves more explanations. In both YIG and GGG the long-range dipole interaction dominates the spin wave dispersion. Their frequencies are governed by the dipolar-gap plus the Zeeman energy ($\hbar f = g\mu_B \sqrt{B(B + 4\pi M_{GGG}(1 - \exp(-kt_{GGG}))/kt_{GGG})}$) as discussed below. Figures R2a and R2b respectively show the spin-wave dispersions at $T = 5$ K at several magnetic fields and the spin-wave gap (at $k = 0$) as a function of B (dashed lines indicate $hf = k_B T$); below (above) these lines, thermal occupation is large (small). In the present nonlocal configuration the wave vector \mathbf{k} is parallel to the magnetic field \mathbf{B} and transport is mainly carried by backward volume magnetostatic waves (BVMSW).

In GGG the frequency f decreases with increasing wave number k and saturates at a constant value for $k > 10^3$ rad/cm. While spin waves with frequencies up to the spin accumulation induced by the spin Hall effect (SHE) in the injector Pt wire are excited, only the thermally occupied ones contribute to long-range transport. The spin current in GGG is therefore conveyed only by the spin waves with significant group velocities close to $k = 0$ ($k < 10^3$ rad/cm). Their frequencies are comparable to the frequency gap determined by the dipole and Zeeman interactions: For $T = 5$ K, the relevant spin-wave frequencies are from ~ 28 to 37 GHz at $B = 1$ T, from ~ 97 to 108 GHz at $B = 3.5$ T, and from ~ 224 to 234 GHz at $B = 8$ T.

The exchange stiffness in YIG (roughly) adds a parabolic contribution to the dipolar dispersion. Figure R2c compares the spin-wave band structures for YIG (red solid lines) and GGG (blue solid lines) at $T = 5$ K and several B , where the dashed lines indicate again $hf = k_B T$. At high temperatures, exchange spin waves with high group velocities are excited in YIG (the $\sim k^2$ term). Even at low temperatures, exchange waves play a role in YIG because they cause the up bending of the flat dipolar bands and generate a band minimum.

The frequency of the dominating spin waves in nonlocal transport is often referred to as $hf = k_B T$, but this is not precise. Spin waves (magnons) obey Bose-Einstein statistics for a certain temperature T and chemical potential μ , and the scattering times are frequency

dependent. The expression of “spin waves with $hf = k_B T$ ” rather emphasizes that the relevant spin waves (magnons) are thermally populated and at elevated temperatures far from the spin wave band minimum.

To clarify these points, we revised the discussion (Lines 155 - 167, Pages 11 - 12).

Figure R2: Computed spin wave dispersions and gaps of GGG and YIG. a, Spin-wave dispersions of GGG (thickness = 500 μm) at $T = 5$ K at $B = 1, 3.5,$ and 8 T for $k < 5 \times 10^3$ rad/cm. **b,** The spin-wave excitation gap of GGG at $k = 0$ at $T = 5$ K as a function of B . **c,** Comparison between the spin-wave dispersions of GGG (blue solid lines) and YIG (red solid lines, thickness = 5 μm) at $T = 5$ K at $B = 1, 3.5,$ and 8 T for $k < 5 \times 10^6$ rad/cm. The dashed lines indicate $hf = k_B T$. The spin-wave dispersions were calculated following Serga *et al.* J. Phys. D: Appl. Phys. **43**, 264002 (2010), with spin-wave propagation (wave vector) \mathbf{k} parallel to the magnetic field \mathbf{B} .

3. Field dependence of the spin diffusion length should be presented. For YIG, the spin diffusion length decreases for increasing the magnetic field (Figure 5a, PRB 93, 020403(R) (2016)). How about GGG in this work? The authors can use the same method to get the spin diffusion lengths for different magnetic fields easily.

The estimated B dependence of the spin diffusion length λ_{GGG} of GGG at 5 K (see below) is shown in Fig. R3. λ_{GGG} increases with B up to a broad maximum of 1.8 μm at around 2 - 4 T. For $B > 4$ T, λ_{GGG} gradually decreases, consistent with a previous report on YIG [PRB **93**, 020403(R) (2016)]. The low- B behavior ($0 < B < 2$ T) suggests that the B -induced magnetization and the resultant dipole interaction improve spin diffusion.

The revised SI (Section G) contains now a section on the B dependence of λ_{GGG} .

Figure R3: B dependence of λ_{GGG} . λ_{GGG} is estimated by the fit of $V_{\text{nl}}(d) = CK_0(d/\lambda_{\text{GGG}})$ to the d dependence of V_{nl} . The error bars represent the 68 % confidence level (\pm s.d.).

The results in Fig. R3 are obtained from the V vs B results for $d = 0.5, 1.0,$ and $3.0 \mu\text{m}$ for which we carried out highly resolved B dependence measurements at 5 K. The number of samples used for the fitting is small compared to the λ_{GGG} estimates at constant field $B = 3.5 \text{ T}$ in the main text, but we confirmed that results are almost the same for 3.5 T ($\lambda_{\text{GGG}} \sim 1.8 \mu\text{m}$), which justifies our procedure.

4. In the abstract the authors said that at 5K spin conductivity is larger than YIG. That only happens when a large magnetic field is applied. So external magnetic field (4T?) should be mentioned in the abstract, or it will be a claim that is easily misunderstood to the readers.

We agree with the reviewer that the role of the field can be made more explicit. We therefore added the pertinent magnetic field value when quoting the spin conductivity of GGG in the abstract (Line 18, Page 1).

5. Why the spin conductivity of GGG in that condition (5K, 4T) is better than in

YIG?

We attribute the high conductivity to the exceptional quality of the GGG crystal, to the long-range nature of the dipole interaction, and to the large spin ($S = 7/2$). In conventional magnets, such as YIG, at elevated temperatures the short-range (nearest and next nearest neighbor) exchange interaction drives spin transport. The spin waves are then sensitive to the local disorder, such as defects, impurities, and magnetic grain boundaries. In paramagnetic GGG, exchange interaction is negligibly small. A moderate magnetic field generates a substantial magnetization by polarizing the large spins. The associated magneto-dipolar interaction generates a stiffness that can support collective spin-wave excitations that carry a spin current. The long-range dipolar interaction is not affected by (short-range) disorder, which is effectively averaged out over typical wave lengths. Therefore, the low-temperature, high magnetic field spin conductivity in paramagnetic GGG can be larger than that of YIG.

We reconstructed the discussion part and added a new paragraph devoted to our interpretation of the largest spin conductivity of GGG (Lines 149 - 167, Pages 11 - 12).

6. Also the authors compared their spin conductivity results of GGG to the values in PRB 94, 180402(R) (2016) where the thickness of YIG is 210nm, with 500 μ m GGG on the bottom. Do the authors think the GGG substrate of YIG plays a role in the transport?

GGG should be inert in the experiments on YIG/GGG at magnetic fields of only 10 mT [Phys. Rev. B **94**, 180402(R) (2016)], which are not high enough to polarize the paramagnet.

To clarify this point we added a paragraph to the discussion in the revised manuscript (Page 10, Lines 130 - 134). Besides, in our answer to Reviewer #1's Query 3, we also discuss conditions at which the GGG substrate do affect the measurements

7. The authors emphasize that the spin transport in GGG persists up to temperatures much higher than GGG's glass-like transition temperature. What are the expectations of the authors for the GGG spin transport measurements

around or below the spin glass temperature?

We thank the reviewer for suggesting an interesting experiment: nonlocal spin transport in a spin-glass phase. Unfortunately, we find it difficult to predict with confidence the spin transport in spin-glass systems. In general, the coherence length for spin excitations becomes shorter when entering the spin-glass phase since the scattering by orientational spin disorder suppresses transport below the spin glass temperature T_g . On the other hand, in the dipolar spin wave regime the disorder should not be so important, as argued above and consistent with the findings of Ochoa *et al.* [Phys. Rev. B **98**, 054424 (2018)], who even predict that a spin glass is beneficial for spin superconductivity. Future low-temperature experiments around the T_g of GGG, should help to elucidate this issue. In the revised manuscript, we mentioned this perspective in the conclusions (Lines 171 - 173, Page12).

8. What is the difference of the experiment data between figure 2d, figure 3b right panel, figure 3d 5 K and figure 4b?

We apologize for the unclear figure captions. The original data for Figs. 2c, 3b (right panel), 3d (for 5 K), and 4b are the same: the B dependence of V in a $d = 0.5 \mu\text{m}$ Pt/GGG/Pt device with $I = 100 \mu\text{A}$ and $f = 3.423 \text{ Hz}$ at $T = 5 \text{ K}$.

We use the same data repeatedly for a better step-by-step explanation of our results. Figure 2c shows $V(B)$ for $|B| < 4 \text{ T}$, highlighting the observation of the nonlocal signal in the Pt/GGG/Pt and the trend, i.e., V increases with B up to $\pm 4 \text{ T}$. Fig. 3 then focuses on the signal dependence on high-magnetic field and temperature. Figure 4 compares our theoretical model and the experimental results. To this end, the same V - B measurement data are replotted in Fig. 4b.

To clarify this point, we revised the captions in Figs. 2, 3, and 4.

9. Figure 2 caption “ \mathbf{B} and \mathbf{E}_{ISHE} denote the directions of the applied magnetic field and the electric field induced by the ISHE, respectively”. \mathbf{E}_{ISHE} is not labeled in Figure 2b.

We thank the reviewer for spotting this issue. We added the label \mathbf{E}_{ISHE} at the detector Pt wire in Fig. 2b.

Reviewer #3 (Remarks to the Author):

This manuscript reports the highly efficient spin transport in the paramagnetic insulator Gd₃Ga₅O₁₂ (GGG). By using the nonlocal geometry of Pt/GGG/Pt, the authors have found magnon transport in range of micrometer scale at temperatures far above the magnetic glass-like transition temperature. They have determined the characteristic spin transport parameters. Remarkably, at 5K the spin diffusion length reaches 1.8 micrometers; and spin conductivity is even much larger than that of the best YIG, so is the spin mixing conductance at the interface. The authors have proposed that the spin transport mechanism in GGG is mainly due to the long-range dipolar interaction, whereas the magnon excitation in this paramagnetic insulator is caused by the process of interfacial spin-flip. The results are quite interesting in physics. I recommend its acceptance by Nature Communications. But the authors may answer the following remarks to clarify some issues.

We thank the reviewer for appreciating our work and recommending publication in Nature Communications. In the following, we will reply to his/her questions and comments.

1. The earlier nonlocal magnon spin transport experiments are mainly based on the structure of GGG (substrate)/YIG/Pt. Since the authors have claimed that the spin conductivity in GGG is even larger than that of YIG at low temperature (e.g. 5 K), is the early results about YIG spin conductivity and spin diffusion length (e.g. ref [10]) possibly contaminated because of the contribution of GGG at low temperature? It is better for the authors to give some comments about this issue.

We thank the reviewer for bringing up this point, also raised by the other referees, see especially our response to Reviewer #1's Query 3. We believe that the spin conductivity and spin diffusion length of YIG in the early work [Cornelissen *et al.* Phys. Rev. B **94**, 180402(R) (2016)] are not contaminated by the GGG substrate, since their low-temperature experiments were carried out at weak fields of ~ 10 mT, which are not

sufficient to magnetize and activate GGG.

In the revised manuscript, we have added a paragraph to address this point (Page 10, Lines 130 - 134).

2. The authors had better show the second-harmonic signal results, which are often considered in dealing with paramagnetic spin Seebeck effect. Magnon can be excited in magnetic insulators in a variety of ways, such as thermal gradient or low-frequency currents in heavy metals like Pt. It was reported that paramagnons and short range magnetic interaction are important in the thermal induced magnon excitations and diffusion in GGG (ref [17]), which seems contrary to the present conclusion that long-range dipolar interaction plays an important role. The authors should give some comment on this issue.

[Redacted]

REVIEWERS' COMMENTS:

Reviewer #1 (Remarks to the Author):

I thank the authors for the answers and explanations. I think the modified manuscript has the potential for generating further activities in spintronics/magnonics and I am happy to recommend publication.

Reviewer #2 (Remarks to the Author):

From my point of view the authors took all the comments and suggestions from the referees and answered their points in a good way.

To support the discussion part, Figure R2 (computed spin wave dispersions and gaps of GGG and YIG) of the authors' response letter is needed to put in the supplementary material.

I think the general improvement of the content is worth publication on nature communications.

Line 181 in supplementary information:

"On the other hand, g_s in Fig. 6b depends only weakly on the magnetic field." Fig. S6b.

Reviewer #3 (Remarks to the Author):

The authors have improved the manuscript by considering my comments (some remarks are similar for me and other referees). By systematically comparing the difference between the thermo-induced spin current [Phys. Rev. Lett. 114, 186602 (2015)] and current-induced nonlocal spin current in GGG, as well as classifying the different spin current transport mechanisms in YIG [Nat. Phys. 11, 1022 (2015); Phys. Rev. B 94, 180402(R) (2016)] and GGG, it's confirmative to say that paramagnets represented by GGG possess unique features for spin transport. Therefore, I would like to recommend it for publication in Nature Communications.

Author's response to Reviewers

Reviewer #1 (Remarks to the Author):

I thank the authors for the answers and explanations. I think the modified manuscript has the potential for generating further activities in spintronics/magnonics and I am happy to recommend publication.

We thank the reviewer for appreciating our revised manuscript and recommending for publication in Nature Communications.

Reviewer #2 (Remarks to the Author):

From my point of view the authors took all the comments and suggestions from the referees and answered their points in a good way.

To support the discussion part, Figure R2 (computed spin wave dispersions and gaps of GGG and YIG) of the authors' response letter is needed to put in the supplementary material.

I think the general improvement of the content is worth publication on nature communications.

Line 181 in supplementary information:

"On the other hand, g_s in Fig. 6b depends only weakly on the magnetic field." Fig. S6b.

We thank the reviewer for her/his valuable time spent on the evaluation of our manuscript and recommendation for publication in Nature Communications. Following the suggestion, we added Supplementary Discussion devoted for explaining the spin-wave dispersion and frequency of GGG and YIG including Figure R2. We also thank the reviewer for pointing the typo.

Reviewer #3 (Remarks to the Author):

The authors have improved the manuscript by considering my comments (some

remarks are similar for me and other referees). By systematically comparing the difference between the thermo-induced spin current [Phys. Rev. Lett. 114, 186602 (2015)] and current-induced nonlocal spin current in GGG, as well as classifying the different spin current transport mechanisms in YIG [Nat. Phys. 11, 1022 (2015); Phys. Rev. B 94, 180402(R) (2016)] and GGG, it's confirmative to say that paramagnets represented by GGG possess unique features for spin transport. Therefore, I would like to recommend it for publication in Nature Communications.

We thank the reviewer for her/his appreciation of our work and recommendation for publication in Nature Communications.